# Single-Atom-Based Catalysts for Photocatalytic Water Splitting on TiO$_2$ Nanostructures

Seyedsina Hejazi [1], Manuela S. Killian [1,*], Anca Mazare [2,3,*] and Shiva Mohajernia [1,*]

1   Department of Chemistry and Biology, Chemistry and Structure of Novel Materials, University of Siegen, Paul-Bonatz-Str. 9-11, 57076 Siegen, Germany
2   Department of Materials Science and Engineering, Institute for Surface Science and Corrosion, Friedrich-Alexander-Universität ErlangenNürnberg, 91058 Erlangen, Germany
3   Advanced Institute for Materials Research (AIMR), National University Corporation Tohoku University (TU), 2-1-1 Katahira, Aoba-ku, Sendai 980-8577, Japan
*   Correspondence: manuela.killian@uni-siegen.de (M.S.K.); anca.mazare@fau.de (A.M.); shiva.mohajernia@uni-siegen.de (S.M.)

**Abstract:** H$_2$ generation from photocatalytic water splitting is one of the most promising approaches to producing cost-effective and sustainable fuel. Nanostructured TiO$_2$ is a highly stable and efficient semiconductor photocatalyst for this purpose. The main drawback of TiO$_2$ as a photocatalyst is the sluggish charge transfer on the surface of TiO$_2$ that can be tackled to a great extent by the use of platinum group materials (PGM) as co-catalysts. However, the scarcity and high cost of the PGMs is one of the issues that prevent the widespread use of TiO$_2$/PGM systems for photocatalytic H$_2$ generation. Single-atom catalysts which are currently the frontline in the catalysis field can be a favorable path to overcome the scarcity and further advance the use of noble metals. More importantly, single-atom (SA) catalysts simultaneously have the advantage of homogenous and heterogeneous catalysts. This mini-review specifically focuses on the single atom decoration of TiO$_2$ nanostructures for photocatalytic water splitting. The latest progress in fabrication, characterization, and application of single-atoms in photocatalytic H$_2$ generation on TiO$_2$ is reviewed.

**Keywords:** titanium dioxide; water-splitting; H$_2$ generation; single-atom



## 1. Introduction

Photocatalytic water splitting is the process of H$_2$ and O$_2$ production from water under light irradiation and in the presence of a catalyst, and follows the Equations (1)–(3) listed below.

$$2H^+ + 2e^- \rightarrow H_2 \tag{1}$$

$$2H_2O + 2h^+ \rightarrow \frac{1}{2}O_2 + 2H^+ \tag{2}$$

$$H_2O \rightarrow H_2 + \frac{1}{2}O_2 \tag{3}$$

Basically, in the photocatalytic pathway, photon energy (from solar energy) converts to chemical energy (hydrogen) in the presence of photocatalysts. The ground-breaking work of Fujishima and Honda [1] demonstrated the feasibility of photocatalytic hydrogen generation from water, using Pt/TiO$_2$ electrodes.

Currently, hydrogen is produced mainly from fossil fuels with a rate of ~50 billion kg per year [2]. Thus, solar water splitting can become a sustainable and renewable hydrogen production path, with the potential to mitigate the energy crisis.

Semiconductors are among the most suitable materials as photocatalysts for H$_2$ production from water splitting or photo-reforming of H$_2$O/alcohol mixtures, e.g., using methanol or ethanol (without or with the presence of co-catalysts). According to Takanabe [3], the process of water splitting on a semiconductor involves six major components

that occur sequentially (see Figure 1), which schematically summarizes the contribution of all the steps influencing the overall water splitting. In the first step, upon irradiation, electron-hole pairs are generated. This occurs when the photon energy is larger than the bandgap of the semiconductor. The second step is charge separation, followed by the third step of diffusion of the photogenerated charge carriers. Carrier diffusion (the third step) depends mainly on carrier lifetime, mobility, and diffusion length. Electrons and holes recombination is almost zero in an ideal photocatalyst with maximum efficiency. The fourth step is carrier transportation and is confined by the following characteristics of the system, conductivity, space charge layer, and flat band potential. The fifth major step is the catalyst's efficiency in terms of exchange current density, Tafel slope, electrocatalytic activity, activation energy, and charge transfer resistance. In the last step, mass transfer plays an important role and is under the control of mainly the iR drop in the solution, pH gradient, viscosity, and effective size of the ions in the electrolyte.

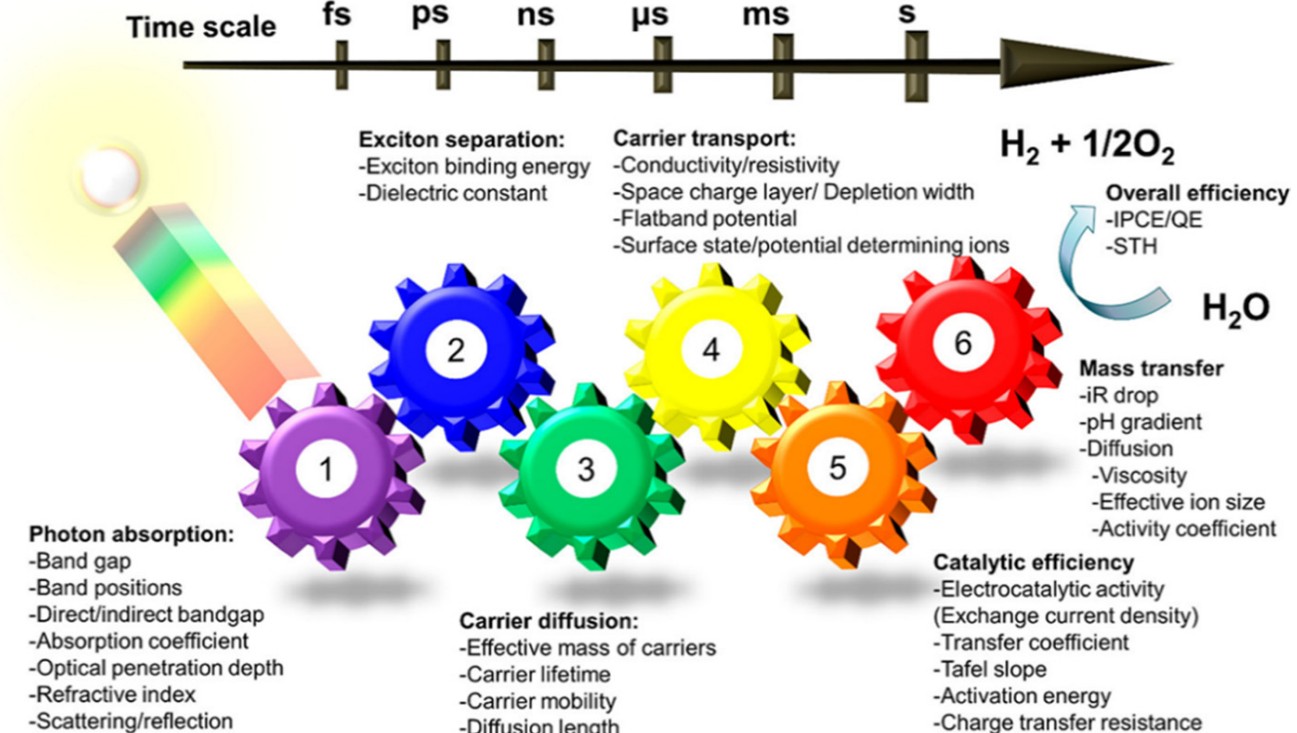

**Figure 1.** The parameters affecting the overall photocatalytic water splitting. The gear with the number indicates the order of the photocatalytic process to be successful for overall water splitting. The timescale for the reactions is also shown. Reprinted with permission from Ref [3], Copyright © 2017 American Chemical Society.

There are excellent review articles with respect to photocatalyst and photoelectrochemical water splitting, for example, refs. [4–10]. Of the multitude of investigated semiconductor materials, $TiO_2$ remains one of the most investigated photocatalysts as it provides excellent photo corrosion, non-toxicity, a low cost, and, most importantly, the conduction and valence band edges [11,12]. However, the solar to hydrogen (STH) conversion efficiency is still very low for $TiO_2$, and the insufficient energy conversion efficiency in $TiO_2$ systems is mainly ascribed to the following factors: (i) the fast recombination of photogenerated charge carriers (electron-hole pairs), the recombination of electrons on the conduction band with the holes in the valence band causes unwanted heat or photon generation; (ii) the backward reaction, the reaction of $H_2$ and $O_2$ to form water, which occurs very fast; and (iii) the wide band gap of $TiO_2$ (3.2 eV) that allows only UV light absorption for $H_2$ production. Since only about 4% of the solar radiation energy is UV light the STH efficiency remains very low with pristine $TiO_2$ structures.

In general, rutile TiO$_2$ is the most stable phase among the TiO$_2$ polymorphs, compared to anatase and brookite, and has a smaller band gap. However, anatase is the preferred polymorph in photocatalytic applications due to a higher conduction band energy and a lower electron-hole recombination rate [12,13]. Different shapes of TiO$_2$ nanostructures have been synthesized to modify the electronic and optical properties, e.g., nanoparticles, nanobelts, nanosheets, nanoflakes, nanotubes [11]. From the different methods, the electrochemical anodization of Ti foil is a straightforward method to produce 1D nanostructures, and hydrothermal techniques as well provide scalable and convenient methods to produce 1D and 2D nanostructures. Additionally, designing anatase nanocrystals with 001 facet as the most active site using hydrothermal method is of great interest for enhancing the catalytic activities [14].

To tackle the above-mentioned issues and make photocatalytic water splitting feasible, tremendous efforts have been made to enhance the photocatalytic activity and visible light absorptions of TiO$_2$. The following strategies prove to be the most efficient approaches to improving photocatalytic activity: (a) addition of hole scavengers (electron donors) [15–18], (b) addition of carbonate salts [19,20], (c) metal and non-metal doping [21], (d) self-doping (defect engineering) [22,23], and e) noble metal co-catalyst decoration [24]. Among the mentioned strategies, the use of noble metals as a co-catalyst for photocatalytic water splitting is one of the most efficient strategies. Platinum group metals (PGM) such as Pt, Pd, Rh, Ru, or Ir are among the least abundant elements on the earth and are thus expensive. However, they are the best-known catalysts for photocatalytic H$_2$ evolution [25]. Therefore, there is significant interest in utilizing noble metal catalysts in heterogeneous photocatalysis. For metal particles, the ultimate size limit is the single-atom catalyst, i.e., isolated atoms dispersed on supports. In addition to their small size that leads to a higher active surface area, a characteristic is also a low coordination environment that is ascribed to unsaturated bonding on the single-atoms [26,27]. Additionally, single-atoms have a distinctive HOMO–LUMO gap due to quantum size effects [28]. Furthermore, the chemical bonding between single-atom metals and supports facilitates the charge transfer between the single-atom co-catalysts and supports [29,30].

It should be noted that the development of the single-atom catalysis field largely relies on advances in both theoretical modeling and atomic-resolution characterization techniques. The present review is specifically focused on TiO$_2$ nanostructures for photocatalytic water splitting, using single-atom noble metals or transition metals as co-catalysts.

## 2. Fabrication of Single-Atom Co-Catalysts

Due to their small size, single atoms have high surface energy; thus, they often tend to agglomerate during the reaction process. One of the main goals in single-atom research is overcoming aggregation, which requires the formation of strong bonding between the single-atoms and the support material. This interaction directly influences the stability and efficiency of the reaction for which a single atom is adopted. The key to forming stable single-atom catalysts with high density is to enhance the anchoring sites. There are three types of anchoring sites for fabricating single-atom catalysts: (i) defects on the support material [31], (ii) unsaturated coordinated atoms [31], and (iii) excess O atoms on the support surface to form hollow sites [31]. Figure 2 illustrates the paths and key strategies for single-atom decoration [32], which will also be further discussed below.

In general, there are two main strategies for constructing single-atom catalysts with well-defined and separated atomistic structures, namely high vacuum physical deposition and wet chemical deposition

### 2.1. High Vacuum Physical Deposition

Vacuum deposition is a more suitable technique for fundamental studies mainly due to the superior control over the single-atom deposition and model catalyst formation. However, as a result of the low yield and high cost of this method, it is not yet feasible for

commercial catalyst production [33,34]. Mass-selected soft-landing [35] and atomic layer deposition (ALD) [36] techniques were further developed for the fabrication of single-atom catalysts. Mass-selected soft landing (Figure 2c) includes an ion source in gas form and a mass spectrometer to filter the nanoclusters according to their mass before their deposition onto a substrate [33].

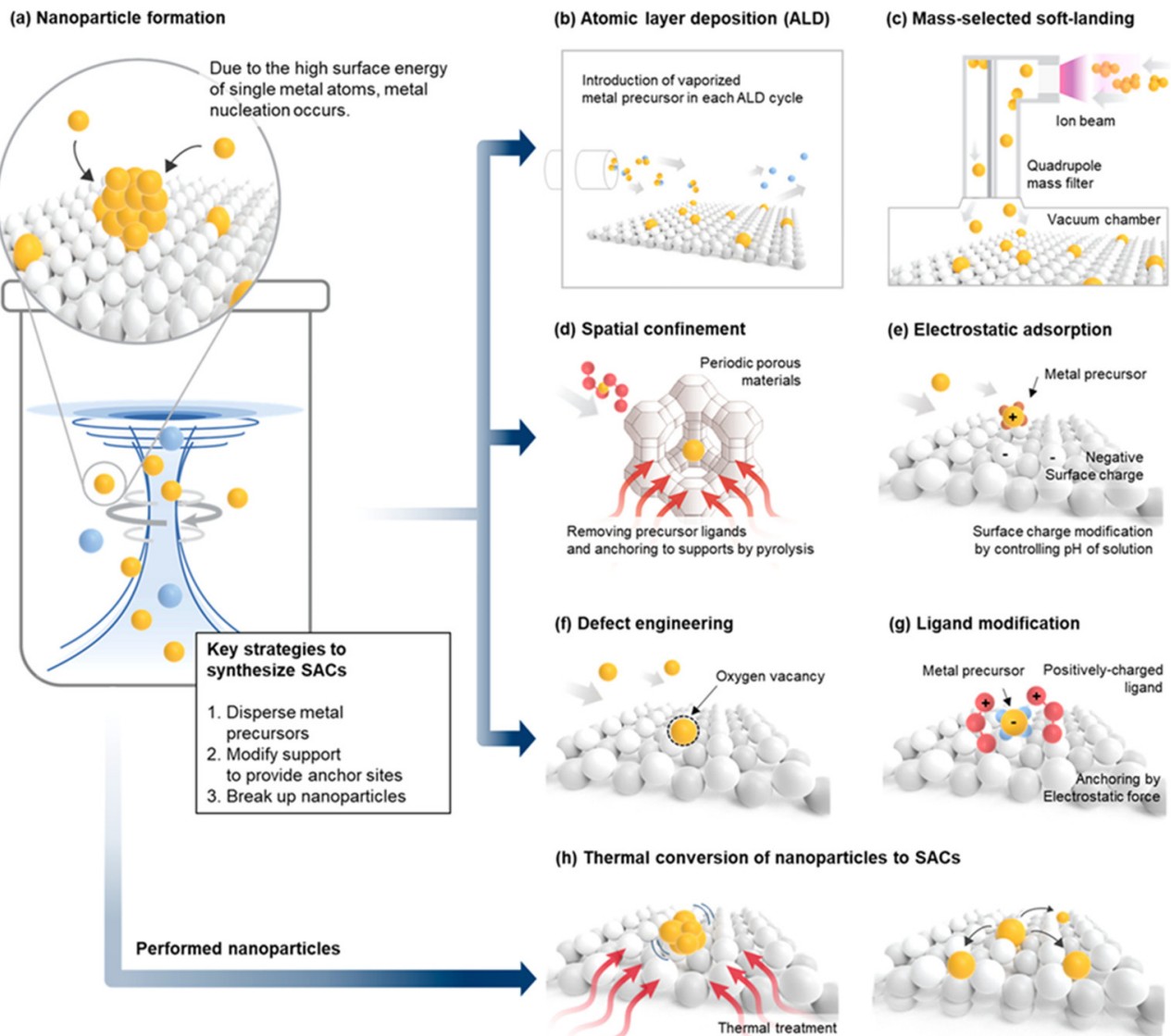

**Figure 2.** Schematic illustration of single-atom catalysts fabrication methods. Reprinted with permission from Ref [32], Copyright © 2020 American Chemical Society.

As a result, metal clusters with a well-defined number of atoms can be deposited (soft-landed) on a substrate. Kaden et al. [37] reported on the deposition of Pd clusters (Pd(n), for n = 1, 2, 4, 7, 10, 16, 20, and 25) on clean rutile $TiO_2$ (110) to build a model catalyst for CO oxidation.

In ALD synthesis (shown schematically in Figure 2b), precursor molecules chemisorb on the support surface via gas-solid reactions and, in each reaction step, form less than a monolayer on the surface. Repeating this cycle, and depending on both the support and ALD-grown material, it is possible to form thin films, nanoparticles, and even single-atoms [38,39]. By optimizing the ALD conditions, namely temperature, counter-reactant, precursor, and pulse times, as well as selecting appropriate ALD approaches such as low-

temperature selective ALD [40] or modifying the pulsing sequences [41], one can design single-atom catalysts for a wide range of catalytic applications [42].

### 2.2. Wet Chemical Deposition

Wet-chemical approaches are more common and used routinely to synthesize single-atom catalysts as they do not require specialized equipment. According to the integration of the components, single-atom wet chemical synthesis is categorized into bottom-up methods, where single-atoms are anchored to the substrate by the reaction of the metal complexes with the anchoring sites on the substrate surfaces, or top-down methods, where metal nanoparticles are decorated on the surface of the support followed by dispersion into single-atoms [43].

In the bottom-up approach (Figure 2d–g), very dilute metal precursors are used as a source of the single-atom where the support material is dipped inside to be decorated by single-atoms and is usually followed by a calcination step, which results in single-atom catalysts on the support surface by a reduction or activation step [44]. Impregnation [45,46], galvanic replacement [47,48], electrostatic adsorption, and photochemical adsorption [49] are among the most common bottom-up methods for single-atom catalyst fabrication.

Nanostructures are turned into smaller pieces in a top-down strategy to obtain desirable characteristics and performance. For single-atom catalyst fabrication, the high-temperature atomic migration method represents a typical top-down approach (Figure 2h). The mechanism behind this method can be explained by Ostwald ripening. This process includes detaching metal atoms from nanoparticles, diffusion of the atoms on the substrate, and attachment to bigger particles [50] or immobilizing in defects [51]. For example, Ag single-atoms were prepared starting from Ag nanoparticles at 400 °C on $MnO_2$-based substrates [52], and Li et al. [51] showed that noble metal nanoparticles can be thermally converted to highly active and stable single-atoms (Pd, Au, Pt) at above 900 °C in an inert environment. Overall, this high-temperature atomic diffusion technique provides thermally stable and high-performance single-atom catalysts.

### 3. Characterization Methods for Single-Atom Co-Catalysts

Despite their extensive investigation, several key points related to the single-atom catalysts' physicochemical properties still remain unanswered or require further investigation, especially in view of understanding and correlation these properties to their photocatalytic efficiency. This includes the reliable evaluation and characterization of the single-atoms in their environment, considering their very low amounts (low weight loadings), strong metal-support interactions, and so on, characteristics which are necessary in order to identify their amount, charge state, electronic structure, atomic configuration, bonding interactions, etc.

The characterization techniques typically employed for nanoparticle characterization (with their corresponding optimization) cannot be used to confirm the presence of single-atom catalysts, though such techniques are useful for substrate characterization or as a negative control. Nowadays, with the progress achieved in characterization techniques, it is actually feasible to observe single metal atoms or sub-nanometric metal clusters of few atoms by means of aberration-corrected electron microscopy [53] or atomic-resolution scanning tunneling microscopy (STM) [54]. Moreover, their loading and coordination environment can be further evaluated by extended X-ray absorption fine-structure spectroscopy (EXAFS), X-ray absorption near-edge spectroscopy (XANES), X-ray absorption spectroscopy (XAS), and ambient pressure X-ray photoelectron spectroscopy (XPS). Recently, focus was also given to a more thorough understanding of these characterization techniques and their application to the single-atom catalysts field, see the comprehensive reviews in refs. [28,32,55–57]. It is also worth mentioning that supported single atoms are usually stabilized by chemical bonding to the inorganic support (e.g., on transition metal oxides, and zeolites), as previously discussed in the fabrication section. Such SA will show a limited geometric transformation under reaction

conditions, however, when supported on organic polymers with functional groups (like amine, carbonyl groups, thiol, etc.) SA may adapt their coordination environment under reaction conditions (as a result of the interaction between the single-atoms with the substrate molecules) [28]. Therefore, next we will discuss and exemplify characterization techniques for single atoms specifically decorated only on inorganic supports.

### 3.1. Transmission Electron Microscopy (TEM) for SA Evaluation

TEM is one of the most direct methods enabling the detailed evaluation of atomic-scale structural information of single-atom catalysts as well as their interaction with the support material. Moreover, aberration-corrected TEM either in both phase contrast TEM or sub-angstrom resolution high-angle annular dark-field STEM (HAADF-STEM) mode, was successful in imaging single-atom catalysts on various supports [58]. The advantage of HAADF-STEM lies in the fact that the imaging is based on Rutherford scattering (i.e., image intensity for the given atoms is proportional to the square of the element's atomic number), and this results in a brighter contrast of heavy metals atoms when on low mass background supports. Current literature includes the identification of single-atom on graphene-based supports (Au [59], Fe [60], Ru [61], etc.), graphitic carbon nitride (Pt [62]), carbon paper (Pt, Ru [63]) or inorganic semiconductors such as cerium oxide (Pt [64,65], Ru [66]), alumina (Pt [67]) or $TiO_2$ (Pt [45,68–75], Pd [71,76–81], Au [71,82,83], Ir [84,85], Rh [86]). This enabled a precise evaluation of the size and distribution of the single metal atoms, as well as their local structural information (metal species on the support) [55]. For example, HAADF-TEM images of typical Pt SAs on different inorganic semiconductors are shown in Figure 3a for Pt SAs obtained by the L-ascorbic acid (AA)-assisted reduction synthesis on $CeO_2$. Namely, Chen et al. [64] confirmed the presence of atomically dispersed Pt catalysts throughout the $CeO_2$ support (on the surface defects), showing also a uniform distribution of the co-catalyst on the support in the elemental mapping (Figure 3b). Similarly, Hejazi et al. [45] have shown that the Pt is present as SA on $TiO_2$ with a uniform distribution, see Figure 3c,d, and such a disperse SA loading was achieved by a dark deposition method based on wet impregnation.

Thus, TEM enables the direct imaging of supported co-catalyst particles and can identify structural and interfacial information of both the co-catalyst metals and substrate support, but only for specific positions on the sample, which means that it can lack a broader context of the investigated sample or in bulk. Therefore, it would prove helpful to combine TEM with other characterization methods to ensure that the observation is valid for the whole of the material. The recent comprehensive reviews of Kottwitz et al. [56] and Tieu et al. [87], detailing the use of advanced electron microscopy in single-atom catalysis, are also recommended for more details.

### 3.2. X-ray Absorption near Edge Structure (XANES) and Extended X-ray Absorption Fine Structure (EXAFS) for SA Evaluation

XANES and EXAFS are characterization techniques reflecting the oscillatory structure of the studied samples in correlation with the X-ray absorption coefficient [57,88]. The binding energy shifts and wiggles of the single metal atoms in XANES spectra represent the near-neighbor distances and their chemical coordination [57]. For example, XANES spectra would indicate complex structures, including the formation of additional bonds (M–N, M–C, etc.) which can disrupt the precise data associated to isolated metal atoms [57]. A significant advantage of these methods is their ability to provide insight into the chemical nature and stabilization structure of single atoms on supports, as well as the investigation of bulk materials. Combined with TEM, this would lead to the overall structure of metal SA compared to local, surface information [55,89]. The absence of metal-metal interactions can be a clear indication of the absence of particles in which, in general, single supported atoms are oxidized, and thus ruling out the presence of rafts/clusters/nanoparticles.

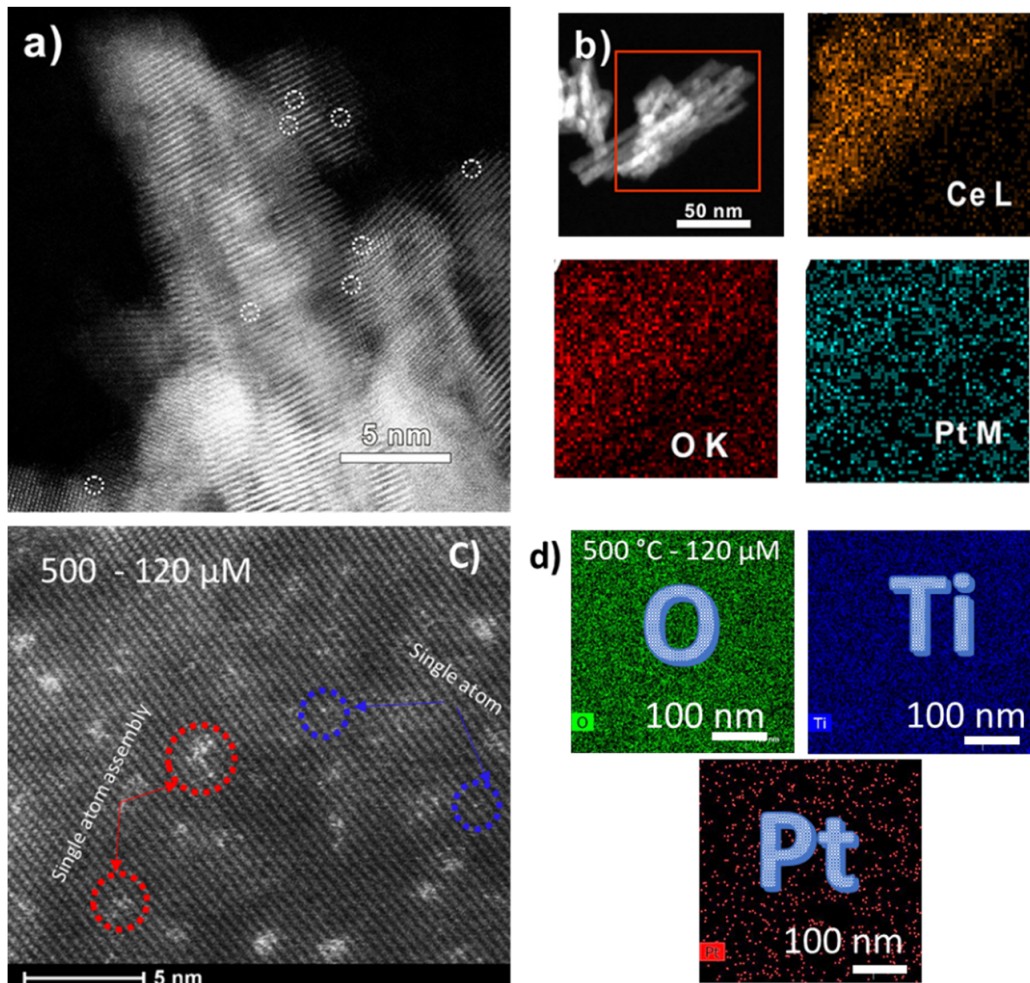

**Figure 3.** Pt on $CeO_2$: (**a**) Cs-corrected HAADF-STEM, (**b**) HAADF-STEM, and corresponding elemental mappings images of $CeO_2$-AA-Pt-cal. (**c**,**d**) Pt on $TiO_2$: (**c**) HAADF-TEM image and (**d**) EDS map of Pt-SA-decorated $TiO_2$ layer. Part (**a**,**b**) is adopted with permission from Ref [64] Copyright © 2018 American Chemical Society. Part (**c**,**d**) is adopted with permission from Ref [45], Copyright © 2020, John Wiley and Sons.

Especially for evaluating the electronic structure of metal SA, XANES measurements are useful for characterizing the electronic and structural properties of the absorbed SA, as shown by Li et al. [90] (Figure 4a). The authors [90] attribute the white line of the Pt L3 edge to the transition of 2p electrons to unoccupied 5d valence orbitals, thus providing information related to the population of unoccupied Pt 5d orbitals. Evaluating the Pt SA on various supports ($Co_3O_4$, $CeO_2$, $ZrO_2$, graphene) and comparing the data with that of Pt foil and $PtO_2$ (as references), resulted in establishing an explicit electronic correlation with the unoccupied Pt 5d states, having the following magnitude for the Pt SAs on the supports $Co_3O_4$ > $ZrO_2$ > $CeO_2$ > graphene [90]. The Pt SAs on the $Co_3O_4$ have shown a higher valence state of ~$4^+$ and a larger population of unoccupied Pt 5d states [90]. Similar investigations were also performed to investigate the Pt 5d occupation state of Pt SA compared to small clusters or nanoparticles (on graphene) [91], for Pt SAs on FeOx [92] or on different polymorphs of $TiO_2$ nanoparticles [93] and for bi-metallic SAs of PtAu [94].

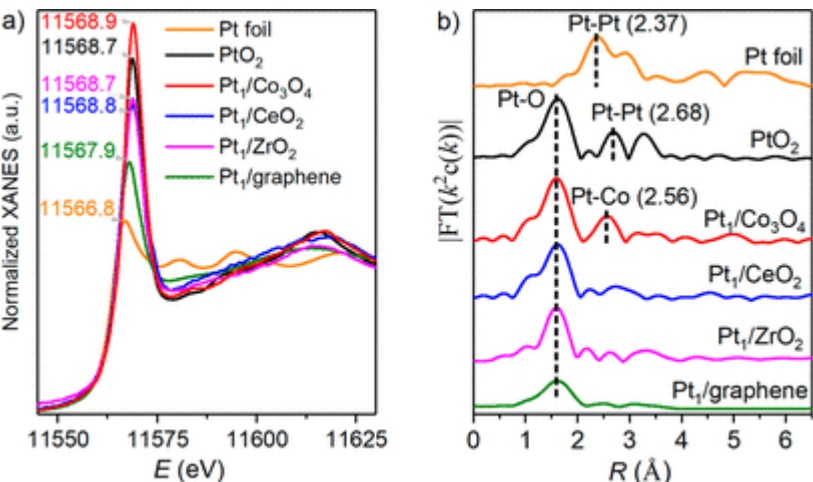

**Figure 4.** (**a**) XANES spectra of Pt1/Co₃O₄, Pt1/CeO₂, Pt1/ZrO₂, and Pt1/graphene single-atom catalysts as well as the Pt foil and PtO₂ reference at the Pt L3-edge. (**b**) The corresponding K2-weighted Fourier transform spectra. With permission from Ref [90], Copyright © 2019 American Chemical Society.

Similarly, Li et al. [90] have further evaluated the EXAFS analysis of the Pt SAs on the various supports, see Figure 4b, and attributed to the coordination number of Pt–O (peak at 1.60 Å) the subsequent trend $Co_3O_4 > CeO_2 > ZrO_2 >$ graphene, for the Pt SAs decorated different substrates, and the absence of the Pt–Pt coordination peak suggested the absence of Pt NPs or clusters. Additionally, the $Pt/Co_3O_4$ curve showed a second shell peak at 2.56 Å (attributed to Pt-Co coordination), distinct from those of Pt foil (2.37 Å) and $PtO_2$ (2.68 Å) [90]. Similarly, the coordination of Pt SAs on different supports ($TiO_2$ [93]) or of other SAs (Au on $TiO_2$ [82]) can also be evaluated. For example Wan et al. [82] reported on the isolated single atomic dispersion of Au on defective $TiO_2$, with an Au–O shell (R = 1.6 Å) and an Au–Ti shell (R = 2.4 Å), without Au–Au and Au–Cl coordination peaks (compared with Au foil and $HAuCl_4$ as reference).

XANES and EXAFS measurements are also advantageous for investigating single-atoms on MOF, zeolites, or graphitic layers, see as examples Ye et al. [95] for Co SAs on MOF (Co–N bonding detected in EXAFS), Yin et al. [96] for Co on N-doped carbon, or Choi et al. [97] for Pt on zeolites.

### 3.3. X-ray Photoelectron Spectroscopy (XPS) for SA Evaluation

XPS is a well-known characterization technique, universally used for evaluating the chemical and compositional properties of surfaces, and more specifically in catalysis of the catalysts and co-catalysts. XPS is a highly surface-sensitive technique ensuring analysis of the top 10 nm of the sample. Moreover, it has shown adaptability and precision to the field of supported metal SA catalysts [56,98], and can also be used in ambient pressure mode allowing for the evaluation of catalysts under the reaction atmosphere (e.g., Simonovis et al. [99], evaluating the behavior of singly dispersed Pt atoms on the surface of Cu(111) in an ambient pressure of CO).

Please note that the trapped metal single atom, denoted as M, shows a chemical shift of $\delta^+$ due to the changed co-coordination (as a result of the oxygen coordination, see the case of Pt SAs on $CeO_2$, where the Pt is bound to oxygen instead of Pt [100]). Compared to classical metallic nanoparticles, noble metal SAs decorated on the same support material show a shift to higher binding energies, which is related to its surface coordination—please see the work of Wan et al. [82] showing a shift in the Au4f XPS spectra of Au SAs (85 eV) compared to Au NPs (84 eV) on $TiO_2$. Moreover, Figure 5a shows a similar shift to higher binding energies for Pt SAs/$TiO_2$, compared to their nanoparticle counterpart. The deconvolution of the Pt4f peak of the Pt SAs (see Figure 5b) results into $Pt^{\delta+}$ and $Pt^{4+}$ (the peaks at 72.93 and 76.28 eV correspond to $Pt^{\delta+}4f_{7/2}$ and $Pt^{\delta+}4f_{5/2}$, whereas the peaks at 75.04 and 78.51 eV can

be attributed to $Pt^{4+}4f_{7/2}$ and $Pt^{4+}4f_{5/2}$), while for the Pt nanoparticle decoration peaks are mainly attributed to metallic Pt (4f7/2 at 71.07eV) [45].

Literature confirms that when Pt is present in the SA state, it is coordinated to its trap environment, that is mainly oxygen atoms of the oxide surface, thus forming Pt-oxide-similar $Pt^{\delta+}$ and $Pt^{4+}$ states (with $Pt^{4+}$ as evidence of a strong Pt–TiO$_2$ interaction) [92,100–103]. Similar results were also obtained for other Pt single-atom decoration on different TiO$_2$ supports [69–71,104] or on other support materials, e.g., Pt on nanosized CeO$_2$ (no metallic Pt, only SA with +2 state) [105]. Moreso, XPS was used to evaluate the chemical state of a variety of other single-atoms co-catalysts, such as Pd [71,78], Au [71], Rh [86,106], and Ir [84], confirming the δ+ state of the single-atom co-catalyst on the evaluated support material.

Recently Wei et al. [94] reported on the synthesis of bi-metallic single-atoms on different facetted TiO$_2$ ({001},{101}), where the metal–support interactions that originated from the coordinatively unsaturated sites contributed to the anchoring of the atomic co-catalysts—namely, onto {001}-TiO$_2$ through Pt–O and Au–O bonds, while on {101}-TiO$_2$ preferentially for Pt–O and Au nanoparticles. This difference was also observed in the high-resolution XPS spectra, as for the Pt4f peaks (Figure 5c), binding energies for Pt4f7/2 at 73.0 (typically attributed to Pt$^{+4}$) are obtained, while for the Au4f peaks (Figure 5d) a shift of 0.2eV to more negative binding energies was observed for PtAu/{101}-TiO$_2$ (where Au is deposited as nanoparticles). The latter may be due to the changes in the binding environment of Au, as the weaker metal-support interactions lead to agglomeration and reduce the binding energies [71,94].

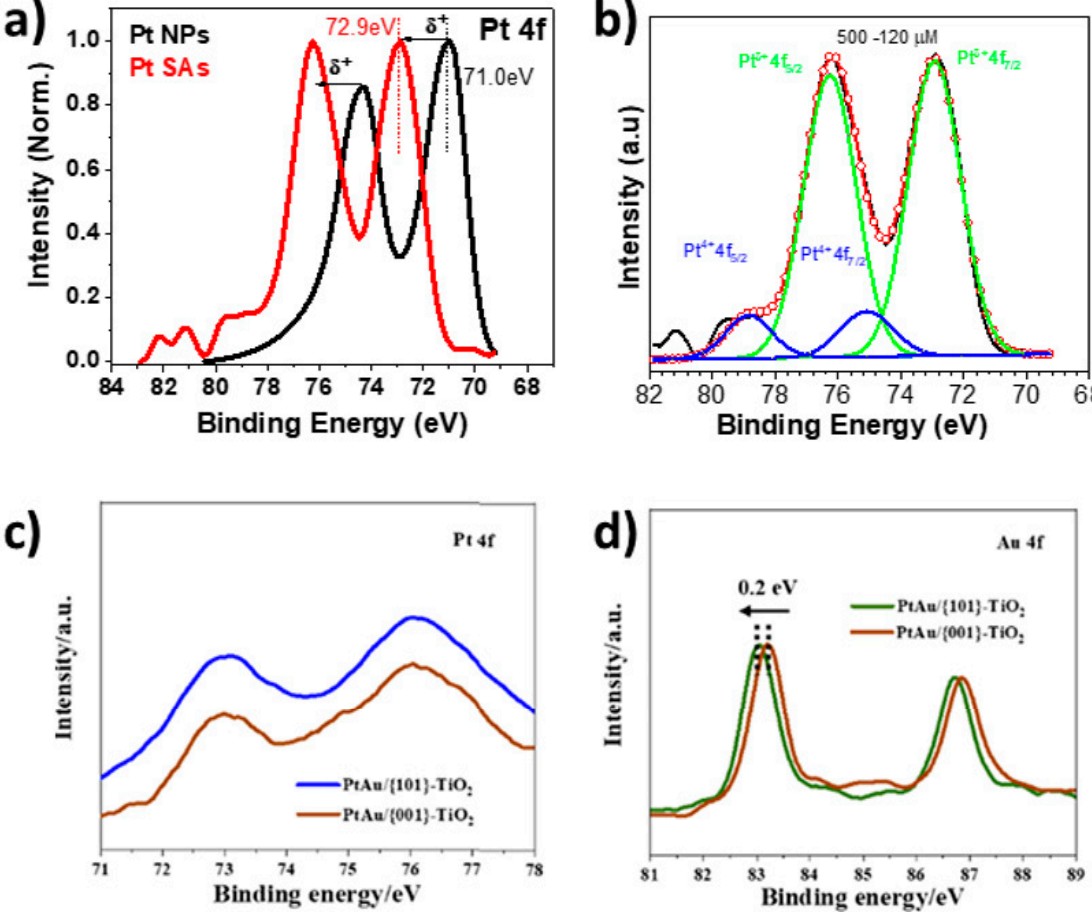

**Figure 5.** (**a**) XPS high-resolution spectra of SAs vs NPs for Pt on TiO$_2$, (**b**) Peak fitting of the Pt SAs of (**a**). Bi-metallic SAs as exemplified for Pt-Au SAs on different facetted TiO$_2$, showing the (**c**) Pt4f and (**d**) Au 4f. Part (**a,b**) reproduced with permission from Ref [45], Copyright © 2020, John Wiley and Sons. Part (**c,d**) reproduced With permission from Ref [94], Copyright © 2021 American Chemical Society.

With respect to the energy shift in XPS, two further aspects need to be mentioned, (i) first that both support and particle size could induce positive binding energy shifts (as shown for Pt nanostructures because of the reduction of effective atomic coordination numbers) [107], and (ii) second, in the case of comparing the different SA binding energies, careful consideration of the peaks used for calibration has to be made (especially if the C1s peak at 284.8eV is employed [108–110]).

As a closing remark, a combination of characterization techniques has to be used to properly identify and quantify the atomic fractions of the SA. Additional characterization techniques adapted to single-atom characterization include (a) Fourier-transform infrared spectroscopy (FTIR)—molecular groups, chemical bonding (only specific molecules can be detected) [98] or diffuse reflectance infrared Fourier transform spectroscopy (DRIFTS)—which provides chemical and structural information related to surface species, commonly through measurement of adsorbed probe (e. g., CO) molecules in studies of catalytic reaction mechanisms, typically linearly adsorbed CO on an isolated metal atom site [56,105], (b) surface-enhanced Raman spectroscopy (SERS)—crystal structure, chemical bonding (though only working for specific elements) [98], (c) electron energy loss spectroscopy (EELS)—element and chemical state analysis [98], scanning tunneling microscopy (STM)—imagining of the supported individual single metal atoms (low temperature under ultrahigh vacuum) [28,98].

## 4. Photocatalytic Hydrogen Evolution on Single Atom Decorated TiO$_2$

Sluggish charge transfer, limited photoconversion, and fast electron-hole recombination rate are the bottlenecks of TiO$_2$ photocatalytic water splitting, and as mentioned earlier, this makes the introduction of a promising co-catalyst essential in order to improve the photocatalytic performance of TiO$_2$. Recently, single-atom catalysts have drawn significant attention due to enhanced active reaction sites, while decreasing the loading compared to conventional nanoparticle catalysts. The active and isolated metal atoms anchored onto the photocatalysts provide more water-molecule adsorption and thus enhance photocatalytic activity. However, the aggregation of SAs during the catalytic reaction is the major issue that leads to a significant drop in the performance of the photocatalysts. Therefore, obtaining a high concentration and highly dispersed single-atom catalysts is the bottleneck in photocatalytic water splitting using SAs. The single-atom catalysts are stabilized by covalent bonding with the carrier or neighboring surface atoms or interionic interaction. In this context, the catalytic activity of a single-atom can be tuned by metal-support interaction, coordination environment, quantum size effect, and so on [111].

Noble metals such as Pt, Au, and Pd are mainly used as co-catalysts in photocatalytic water splitting, and this is due to their efficient charge separation and low activation energy. To reach a more scalable water splitting system, earth-abundant transition metals such as Cu, Ni, Co, and Fe are promising alternatives for noble metals in photocatalysis [112–115]. Utilizing noble metals and transition metals as single atom co-catalysts for producing H$_2$ from water is known to result in higher efficiency and an overall lower cost. In this section, the most recent research on photocatalytic water splitting reactions using single-atom catalysts on TiO$_2$ will be discussed.

Zhang et al. [116] reported on a large loading amount (1.5 wt.% of Cu) and highly dispersed copper single-atom on nanoparticle TiO$_2$ (MIL-125(Ti$_v$)) for photocatalytic H$_2$ generation that exhibits a quantum efficiency of 56% under 365 nm irradiation (see Figure 6). A range of loading from 0.47 wt% to 2.57 wt% of Cu was evaluated in their experiments. The authors attributed this performance to the strongly anchored Cu single atoms on TiO$_2$ that trigger a reversible/self-healing and continuous photocatalytic process that effectively separates photoelectrons to reduce water to H$_2$. The results are supported with various spectroscopic and in-situ experiments as well as with DFT modelling results. Their study reveals the most efficient charge separation by Cu SA and proves the significance of the in-situ self-healing effect of Cu species during the photocatalytic reaction.

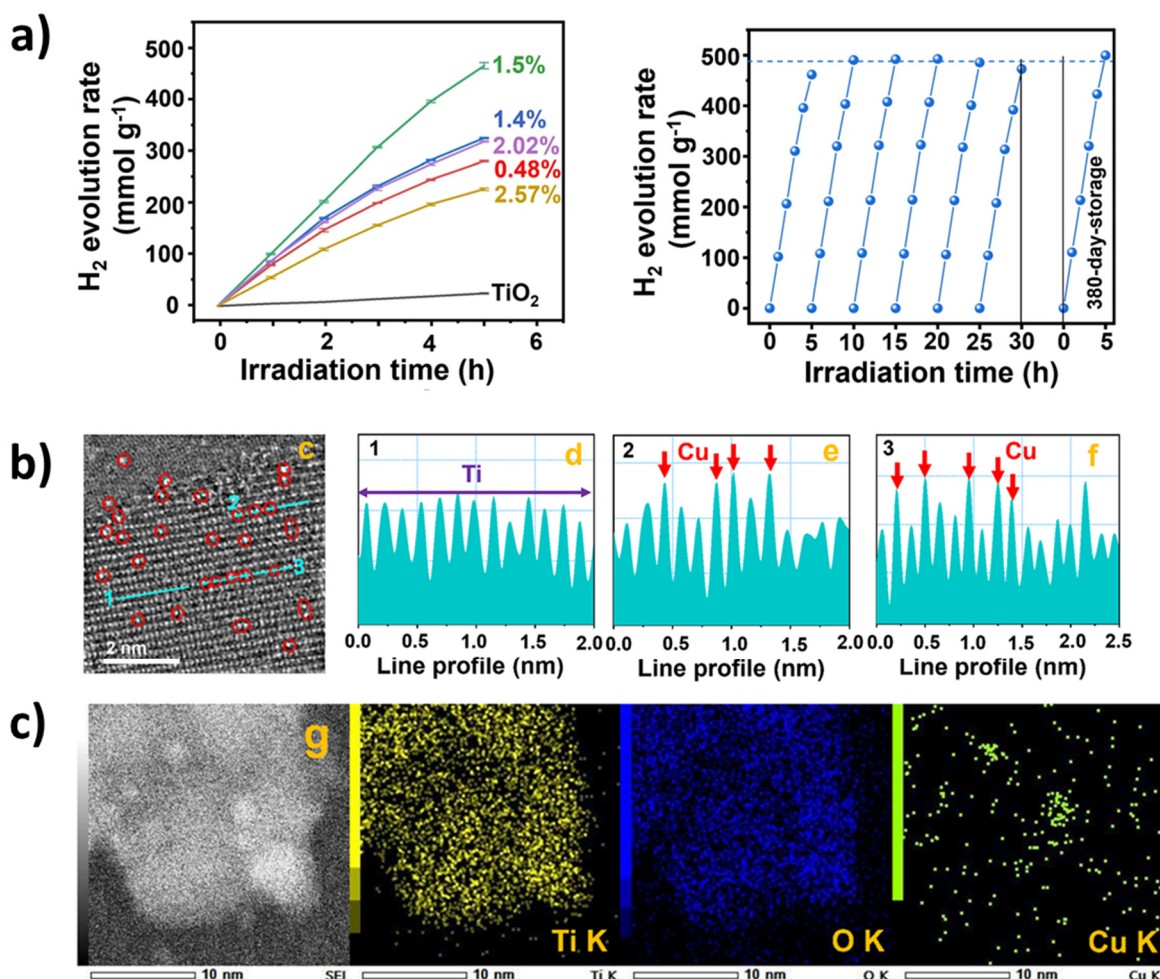

**Figure 6.** (**a**) The photocatalytic $H_2$ evolution rate of $TiO_2$ loaded with different ratios of Cu SA catalysts as a function of the irradiation time (left side) and right side—the photocatalytic activity of the ca. 1.5 wt% CuSA-$TiO_2$ for six cyclic water splitting experiments and the last run is the activity of the sample after storing in the lab for 380 days, (**b**) filtered HAADF STEM image and the corresponding line scan profiles, (**c**) STEM-EDS mapping of Ti, O, and Cu of the fresh CuSA–$TiO_2$. With permission from Ref [116], Copyright © 2021, Springer Nature.

In another approach on $TiO_2$ systems modified by Cu single-atoms, Lee et al. [117] reveal that the cooperative interplay of atomic catalysts and their environment has a significant influence on the properties and catalytic performance of the material. They suggest that the fundamental atomic-level cooperative interaction is very similar to homogeneous catalysts. Through ab initio nonadiabatic molecular dynamics, Long et al. [118] show simulations that Cu single-atoms on $TiO_2$ are inactive for water splitting, before light irradiation. However, after illumination, the deep trap state on the Cud orbital is stabilized by the local distortion around the Cu single atom, decoupling it from the free charge thus providing a high photocatalytic hydrogen production activity.

It is also reported that bi- or multi-metallic catalytic sites show higher photocatalytic water splitting activity compared to regular single-atom catalysts. Wei et al. [94] show that bi-metallic Pt and an Au single atom on the {001}-$TiO_2$ presented a 1000-fold increase in the $H_2$ generation rate in comparison to pristine{001}-$TiO_2$. They show that by regulating the coordination environment of atoms, it is possible to disperse bimetal catalytic sites on faceted semiconductors. They show an $H_2$ evolution rate of 61.3 mmol $h^{-1}g^{-1}$, much higher than that of the $TiO_2$ modified by Pt or Au single-atoms. They explain the mechanism by the synergistic effect of PtAu dual-atom catalysts, which can decrease the Gibbs free energies of hydrogen adsorption by optimizing the electronic states of both Pt and Au sites.

Particularly, the Pt atom is activated by the Au atom and the activity of catalysts is further enhanced through the dimer interaction (see Figure 7).

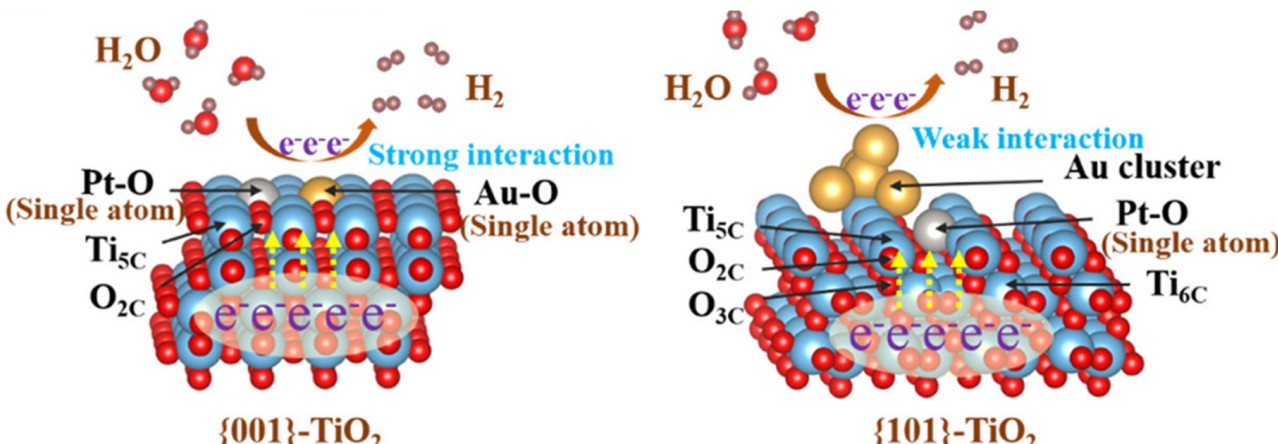

**Figure 7.** Schematic illustration of charge transfer in dual-site co-catalyzed {001} and {101}-faceted $TiO_2$. Blue, red, gray, yellow, and brown spheres represent Ti, O, Pt, Au, and H atoms, respectively. With permission from Ref [94], Copyright © 2021 American Chemical Society.

In another study, Wang et al. [119] reported on the Co and Pt dual single-atom on $TiO_2$ with ultrahigh photocatalytic hydrogen generation efficiency. They attributed this enhancement to the oxygen-coordinated Co–O–Pt dimer coupling, which is able to optimize the electronic states of Pt and Co sites in order to weaken H* binding. They further explain that the Pt single-atom activates the "mute" Co single-atom and the activity of the Pt atom itself is additionally improved by the dimer interaction. With this strategy, the authors show that the dual single-atom catalyst yields 13.4% dimer sites that lead to stable and highly active photocatalysts with an $H_2$ generation rate of 43.467 mmol $g^{-1}$ $h^{-1}$ and external quantum efficiency of ≈83.4% at 365 nm.

Wu et al. [120] reported on the isolated Co atoms on $TiO_2$ nanosheets that significantly improve the charging kinetics via a Co-O electronic coupling. With experimental investigations and theoretical calculations, the authors show that the electron transfer and the hydrogen adsorption/desorption processes are facilitated by the effective atomic scale Co-O electronic coupling.

Several experimental and theoretical studies suggest that coupling Pt single atoms with the atomic scale defects in $TiO_2$ nanostructures is an efficient strategy for stabilizing SAs and to achieving an excellent photocatalytic $H_2$ generation activity compared to conventional nanoparticle decoration [45,68,121,122]. The role of oxygen vacancy in single-atom catalysts expands to Pd for instance, where Wang et al. [77] showed, by experimental and theoretical studies, that the presence of Pd SAs, clusters, and VOs on $TiO_2$ has a synergistic effect on improving the photocatalytic reaction. The authors explain the role of Pd SAs by facilitating the generation and stabilization of oxygen vacancies by the formation of the Pd–O–$Ti^{3+}$ interface, at the same time Pd clusters are responsible for charge carrier separation and act as the optimum active sites for $H_2$ generation, while surface oxygen vacancies are the preferable sites for adsorption and activation/dissociation of reactant molecules. It has also been reported that oxygen vacancies on the surface of $TiO_2$ nanosheets act as traps for Pd single-atoms as cocatalysts for photocatalytic $H_2$ generation [71]. More interestingly, the photocatalytic performance of Pd single-atoms outperforms that of the Pt single-atoms, and according to the DFT calculations, this performance was ascribed to the charge localization on the noble metal single-atoms implanted in the $TiO_2$ surface [71].

Table 1 shows an overview of the photocatalytic $H_2$ evolution rate for various SA co-catalysts on $TiO_2$ support materials discussed above, taking also into consideration the morphology of the support material, the SA loading, light source, and reaction system used. From these data, it is evident that a clear-cut comparison between the different noble

metal single-atom decorations is not feasible, as either there is a significant difference in the noble metal loading amount, the light source used, the difference in the morphology of the support material and the amount of sacrificial agent used in the reaction system.

**Table 1.** Overview of the photocatalytic $H_2$ evolution rate for SA decorated $TiO_2$ nanostructures from literature, including the photocatalytic system (SA metal, SA loading, light source, and reactions system).

| Type of SA | Type of Support | SA Loading | Light Source | $H_2$ Evolution Rate | Reaction System | Reference |
|---|---|---|---|---|---|---|
| Pt | {001} $TiO_2$ nanosheet | 0.47 at% | 325 nm | 3.2 $mL \cdot h^{-1} \cdot cm^{-2}$ | 50 vol% methanol-water | [45] |
| Pt | MIL-125($Ti_v$)(nanoparticle) | - | 300 W Xe lamp | 14 $mmol \cdot h^{-1} \cdot g^{-1}$ | 20 vol% methanol-water | [61] |
| Pd | {001} $TiO_2$ nanosheet | 0.35 at.% | 365 nm | 500 $\mu L \cdot h^{-1}$ | 50 vol% methanol-water | [71] |
| Au | {001} $TiO_2$ nanosheet | 0.35 at.% | 365 nm | 50 $\mu L \cdot h^{-1}$ | 50 vol% methanol-water | [71] |
| Pt | {001} $TiO_2$ nanosheet | 0.35 at.% | 365 nm | 200 $\mu L \cdot h^{-1}$ | 50 vol% methanol-water | [71] |
| Pd | {001} $TiO_2$ nanosheet | 0.39 wt% | Xe lamp | 600 $\mu mol \cdot h^{-1} \cdot g^{-1}$ | N,N-dimethylformamide (DMF) 50, 0.5 mL deionized water and 0.5 mL benzylamine | [77] |
| Pt/Au | {001} $TiO_2$ nanosheet | 0.33 wt% | 300 W Xe lamp | 60 $mmol \cdot h^{-1} \cdot g^{-1}$ | 10 vol% methanol-water | [94] |
| Pt/Au | {101} $TiO_2$ nanosheet | 0.31 wt% | 300 W Xe lamp | 15 $mmol \cdot h^{-1} \cdot g^{-1}$ | 10 vol% methanol-water | [94] |
| Cu | MIL-125($Ti_v$)(nanoparticle) | 1.5 wt% | simulated solar light irradiation | 101.7 $mmol \cdot g^{-1} \cdot h^{-1}$ | $H_2O$/methanol with a ratio of = 1:2 | [116] |
| Cu | $TiO_2$ hallow nanoparticles | 0.75 wt% | Xe lamp | 17 $mmol \cdot h^{-1} \cdot g^{-1}$ | water and methanol (volume ratio of 3:1) | [117] |
| Pt/Co | {001} $TiO_2$ nanosheet | Pt 0.16/Co 0.18 wt% | 300 W Xe lamp | 45 $mmol \cdot h^{-1} \cdot g^{-1}$ | 10 vol% methanol-water | [119] |
| Co | $TiO_2$ nanosheets | 1.11 wt% | 300 W Xe lamp | 8 $mmol \cdot h^{-1} \cdot g^{-1}$ | 1M NaOH solution with 20% methanol | [120] |
| Pt | $TiO_2$ nanobelts | 1 wt% | simulated solar light (AM 1.5G) | 90 $mmol \cdot mg^{-1}$ | 50 vol% methanol-water | [112] |

Using first principle calculations, Yang et al. [123] shed light on the diversity and complexity of the confinement effect of transition metal SA in oxides for $H_2$ evolution. They introduce two stable forms of Co single-atom on the (101)$TiO_2$ surface which are interstices and $Co_1$-substituted $TiO_2$. They prove that the former exhibits better $H_2$ generation activity whereas the latter demonstrates increased hydrogen spillover effect in the $H_2$ atmosphere, producing surface O vacancies and $Ti^{3+}$.

Calculating the free energy changes of hydrogen adsorption in a single-atom Ni/$TiO_2$ system using density functional theory, Bi et al. [124] reveal that the hydrogen generation reactions most probably take place on SA Ni and its surrounding O atoms. They also prove that hydrogen activity in this system is comparable to that of Pt as the best-known catalyst for $H_2$ evolution, and using DFT calculation, identified the active sites on the Ni single atom/$TiO_2$(101) surface at different H coverages.

## 5. Conclusions and Future Outlook

Overcoming the kinetic barrier of photocatalytic water splitting half reactions, which are an $H_2$ evolution reaction and oxygen evolution reaction, is the main challenge for converting solar light to hydrogen. The key to enhance the efficiency of the photocatalytic water splitting reaction is to increase the number and intrinsic activity of the active sites. Remarkably, reducing the size of the photocatalysts simultaneously improves the above-mentioned factors.

In this regard, research on single-atom catalysts has become a frontier in photocatalytic water splitting in the past few years and this is the direct result of the atomistic scale of the single-atoms which leads to a higher active surface area, lower loading amount compared to classical co-catalysts (thus more cost-effective), quantum size effects, facilitated charge transfer and new reaction pathways The advances achieved in both theoretical modeling and atomic-resolution characterization techniques play key roles in the development of single-atom catalysts. For example, the atomic resolution characterization techniques used for the typical nanoparticulate co-catalysts were further developed and tailored to the detection and proper evaluation of the small size of the single atoms. Currently, there is no one characterization technique from which all physico-chemical data of the single-atom loading and their interaction with the surrounding environment can be obtained. There is, however, a mix of techniques which can bring a clear and detailed overview of these properties, and this includes: (i) transmission electron microscopy (HAADF-TEM)—size, distribution, and local structural information of the single metal atoms, combined with additional elemental mapping; (ii) X-ray absorption near edge structure (XANES) and extended X-Ray absorption fine structure (EXAFS)—electronic and structural properties of the absorbed single-atom and their coordination; (iii) X-ray photoelectron spectroscopy (XPS)—coordination state and loading amount of the single-atoms; (iv) or other additional methods such as diffuse reflectance infrared Fourier transform spectroscopy (DRIFTS), scanning tunneling microscopy (STM).

Modification of $TiO_2$, as one of the most studied materials in photocatalysis, with single atoms as cocatalysts for photocatalytic water splitting has become an emerging field of study. The major challenges in single-atom catalysis are connected to the fact that the single-atom loading on the current photocatalysts is very low (<2 wt.%) and to the stability of the SAs, due to their high surface energy. Thus, the focus of research in this area is to enhance the stability and loading of single-atoms on the support material while avoiding the formation of/aggregation to nanoparticles, i.e., obtaining a high coverage of single-atoms on the substrate. In this respect, the interaction of single atoms with the support material has a substantial influence on the stability, loading, and overall photocatalytic activity of the developed SA/$TiO_2$ photocatalytic system. For example, the in situ self-healing effect of Cu species on $TiO_2$ during the photocatalytic reaction resulted in a quantum efficiency of 56% under 365 nm irradiation [116]. Remarkably, the use of bi-metallic single-atoms on a suitable support proves to be an efficient approach in designing a highly efficient photocatalyst. In general, controlling the fabrication of single atoms and operando atomic scale characterizations are the keys to achieving a better understanding of the reactions occurring on single-atoms. Therefore, rapid development in single-atom catalysts for photocatalytic water splitting and, in general, other catalytic applications, is foreseeable.

**Author Contributions:** Writing original draft, review and editing, S.H.; Review and funding acquisition, M.S.K.; Supervision, writing original draft, review and editing, A.M.; Supervision, writing original draft, review and editing, S.M. This manuscript was written by the contributions of all authors. All authors have read and agreed to the published version of the manuscript.

**Funding:** This research was funded by DFG researcher group FOR 1878 and KI 2169/2-1.

**Acknowledgments:** The authors acknowledge the DFG researcher group FOR 1878 and KI 2169/2-1 for funding.

**Conflicts of Interest:** The authors declare no conflict of interest.

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
