# Peer review of "Single-Atom-Based Catalysts for Photocatalytic Water Splitting on TiO2 Nanostructures"

_catalysts, doi:10.3390/catal12080905_

Round 1

Reviewer 1 Report

Overall comment:

Overall, the author presented a mini summary work on single atoms (SA) supported on TiO2 for photocatalytic water splitting. The review includes five parts, background introduction, brief summary of SA preparation strategy, characterization methods for the existence of SA, current advances of photocatalytic water splitting using SA supported on TiO2 and future outlook. The author did a good work in terms of the summary for recent advances on SA supported on TiO2. Rich content is presented. A clean and meaningful review is formulated.

Comment 1: Minor rephrase and format modification for line 207, 214, 239-241, 378 and 388.

Comment 2: Future outlook should include more detailed discussion about the current challenges that single atoms (SA) supported on TiO2 for photocatalytic water splitting shows and future opportunities that exist.

Author Response

Reviewer 1

Overall, the author presented a mini summary work on single atoms (SA) supported on TiO2 for photocatalytic water splitting. The review includes five parts, background introduction, brief summary of SA preparation strategy, characterization methods for the existence of SA, current advances of photocatalytic water splitting using SA supported on TiO2 and future outlook. The author did a good work in terms of the summary for recent advances on SA supported on TiO2. Rich content is presented. A clean and meaningful review is formulated.

Comment 1: Minor rephrase and format modification for line 207, 214, 239-241, 378 and 388.

We thank the reviewer for pointing out these aspects. Corrections were made to the indicated lines and also the whole manuscript was proofread to avoid misspellings.

Comment 2: Future outlook should include more detailed discussion about the current challenges that single atoms (SA) supported on TiO2 for photocatalytic water splitting shows and future opportunities that exist.

We have now modified the future outlook to include the current challenges from SA characterization, their use in water splitting and also for future opportunities.

Reviewer 2 Report

In this review, the author presented the recent and future outlook applications of Nanostructured TiO2 as a highly stable and efficient semiconductor photocatalyst for this H2 generation from photocatalytic water splitting. On the whole, the review is of considerable interest and well done. I recommend it to be published after a minor revision.

1. The introduction is adequate, however, the properties, microstructure and electron energy level structure for TiO2 nanoparticles as a photocatalyst has not been discussed and analyzed in detail, thus, it makes the review article seem shallow.

2. The Authors should also proofread their manuscript (some spelling and grammar errors).

3. The conclusion was not existed; please add it.

4. If possible, some important and relative reports about Photocatalysis over TiO2 could helped: https://doi.org/10.1021/acsomega.1c03693 , https://doi.org/10.1016/j.heliyon.2022.e09652 , https://doi.org/10.1007/s10904-022-02389-8.

Hence, I recommend it accepted for publication after some minor revisions.

Author Response

Reviewer 2

In this review, the author presented the recent and future outlook applications of Nanostructured TiO2 as a highly stable and efficient semiconductor photocatalyst for this H2 generation from photocatalytic water splitting. On the whole, the review is of considerable interest and well done. I recommend it to be published after a minor revision.

  1. The introduction is adequate, however, the properties, microstructure and electron energy level structure for TiO2 nanoparticles as a photocatalyst has not been discussed and analyzed in detail, thus, it makes the review article seem shallow.

We thank the reviewer for pointing this aspect out. We have now included a paragraph related to the different polymorphs of TiO2 and various nanostructures used as photocatalyst.

  1. The Authors should also proofread their manuscript (some spelling and grammar errors).

The manuscript was proofread for spelling and grammar errors, and corrections were made throughout.

  1. The conclusion was not existed; please add it.

We have now modified the last part to a conclusion and future outlook, including now more detailed discussion.

  1. If possible, some important and relative reports about Photocatalysis over TiO2 could helped: https://doi.org/10.1021/acsomega.1c03693 , https://doi.org/10.1016/j.heliyon.2022.e09652 , https://doi.org/10.1007/s10904-022-02389-8.

We thank the reviewer for pointing this out. We have now including the references mentioned by the reviewer which were referring to photocatalysis on TiO2, as well as additional relevant reviews/research works.

Hence, I recommend it accepted for publication after some minor revisions.